# Smart Window with Active-Passive Hybrid Control

**DOI:** 10.3390/ma13184137

**Published:** 2020-09-17

**Authors:** Heng-Yi Tseng, Li-Min Chang, Kuan-Wu Lin, Cheng-Chang Li, Wan-Hsuan Lin, Chun-Ta Wang, Chien-Wen Lin, Shih-Hsien Liu, Tsung-Hsien Lin

**Affiliations:** 1Department of Photonics, National Sun Yat-sen University, Kaohsiung 80424, Taiwan; coololiver0130@gmail.com (H.-Y.T.); book74108520@gmail.com (L.-M.C.); kobe0811219@gmail.com (K.-W.L.); ronnie.ccl8002@gmail.com (C.-C.L.); tiffanywanhsuan@gmail.com (W.-H.L.); wangchunta4@gmail.com (C.-T.W.); 2Material and Chemical Research Laboratories of Industrial Technology Research Institute, Hsinchu 31040, Taiwan; linjianwen@itri.org.tw (C.-W.L.); shliu@itri.org.tw (S.-H.L.)

**Keywords:** smart window, cholesteric liquid crystal, photochromic dichroic dye

## Abstract

Dimming and scattering control are two of the major features of smart windows, which provide adjustable sunlight intensity and protect the privacy of people in a building. A hybrid photo- and electrical-controllable smart window that exploits salt and photochromic dichroic dye-doped cholesteric liquid crystal was developed. The photochromic dichroic dye causes a change in transmittance from high to low upon exposure to sunlight. When the light source is removed, the smart window returns from colored to colorless. The salt-doped cholesteric liquid crystal can be bi-stably switched from transparent into the scattering state by a low-frequency voltage pulse and switched back to its transparent state by a high-frequency voltage pulse. In its operating mode, an LC smart window can be passively dimmed by sunlight and the haze can be actively controlled by applying an electrical field to it; it therefore exhibits four optical states—transparent, scattering, dark clear, and dark opaque. Each state is stable in the absence of an applied voltage. This smart window can automatically dim when the sunlight gets stronger, and according to user needs, actively adjust the haze to achieve privacy protection.

## 1. Introduction

Global warming has become the most important environmental issue. Smart windows are very important in energy-saving buildings, where they save energy and contribute to the reduction of carbon emissions [1,2,3]. A smart window allows the transmission of light to be adjusted; it can protect against sunlight and thermally insulate, effectively reducing the need for lighting and air conditioning. Liquid crystal has been used to manufacture various light shutters and switchable windows. Methods for controlling liquid crystal smart windows are divided into two types—electrical control and photo control. The advantage of an electrical-controllable liquid crystal smart window is that the user can freely set the transmittance or the degree of transmittance by adjusting the applied voltage (see, for example, the dynamic scattering mode [4,5], polymer-dispersed liquid crystals [6,7,8,9], polymer-stabilized cholesteric texture [10,11,12], and dye-doped liquid crystals [9,13]). Since the general nematic liquid crystal has dielectric anisotropy, its liquid molecules rotate in an applied electric field, so electrical control is an ordinary driving method for a liquid crystal smart window. The advantage of a photo-controllable liquid crystal smart window is that it can automatically adjust transmittance from high to low upon exposure to UV, as in sunlight, and so self-adjust without extra energy input. The photo-controllable liquid crystal smart window requires additional photo-sensitive material (such as photosensitive chiral azobenzene [14,15], photochromic dye [16], or azo dye [17]). Most of the photo-controlled liquid crystal smart windows perform the sole function of either absorption or scattering. A single device with multiple states to meet different haze and tint needs is highly desired but challenging to develop. Most liquid crystal smart windows cannot have both photo- and electrical-controllable capabilities, and their transmittance cannot be adjusted to meet user needs.

Cholesteric liquid crystal (CLC) is a smart window material that exhibits bi-stability [18,19]. This bi-stable characteristic eliminates the need for the continuous application of a voltage, as is required with ordinary smart windows. A voltage pulse has only to be applied when switching, so power consumption is very low. CLC has two stable states, which are associated with different optical properties. The first state is the planar state [18], in which CLC virtually functions as a chiral Bragg grating with a photonic bandgap that is centered at wavelength λ_c_ = (n_o_ + n_e_)p/2 and has band width Δλ ≈ (n_o_ – n_e_)p, where n_o_ is the ordinary refractive index, n_e_ is the extraordinary index, and p is the helical pitch. The band gap is easily tunable by adjusting the concentration of the chiral agent. Inside the band gap, the incident light is reflected by the CLC; outside the band gap, the light directly passes through the CLC without scattering. The other stable state is the focal conic state, in which the helical axes are randomly distributed, forming a multi-domain structure with strong light scattering. The CLC must generally have a particular cell gap-to-pitch ratio (d/p) and surface alignment to maintain its stability [13,20]. In a vertical alignment film with large d/p, the cholesteric liquid crystal cannot be easily be put into the planar state, disfavoring transparency; in a homogenous alignment film with small d/p, the cholesteric liquid crystal cannot easily enter the focal conic state, disfavoring scattering and resulting in a poor contrast ratio. In recent researches, a cholesteric liquid crystal light shutter has been developed. This shutter has a cholesteric liquid crystal with negative dielectric anisotropy; it is switched by a high-frequency voltage pulse switched into the transparent planar state and by a low-frequency voltage pulse back into the scattering focal conic state [21,22,23]. This operation increases the tolerance of the CLC for d/p and boundary conditions, and a high-frequency voltage can be used to obtain good planar states even when the cell gap is large or pitch is short (large d/p).

In this work, a photochromic dye is added to a salt-doped cholesteric liquid crystal. The photochromic dye is photo-controllable and automatically darkens when irradiated by UV light. It can regulate the transmittance without sacrificing transparency. The electrical controllability of the CLC is used to control the translucency of smart windows to provide privacy protection. Finally, four optical states are realized—transparent, scattering, dark clear (chromatic and transparent), and dark opaque (chromatic and hazy).

## 2. Materials and Methods 

### 2.1. Preparation of Materials and Measurements

The cholesteric liquid crystal (CLC) mixture was made from 96 wt.% negative dielectric anisotropy nematic liquid crystal DNM-9528 (extraordinary index n_e_ ≈ 1.5792, ordinary index n_o_ ≈ 1.4808, dielectric anisotropy Δε = ε_//_ – ε_⊥_ ≈ –4.8, rotational viscosity coefficient γ1 = 96 mPa∙s, splay elastic constant K11 = 12.4 pN and bend elastic constant K33 = 12.8 pN), doped with 4 wt.% chiral agent R-5011 (HTP ≈ 100 μm^−1^, from HCCH, Jiangsu, China), whose chemical structure is presented in Figure 1; the pitch of the CLC was approximately 250 nm. Then, 97.65 wt.% CLC mixture was further homogeneously blended with 0.35 wt.% tetrabutylammonium tetrafluoroborate ((CH_3_CH_2_CH_2_CH_2_)_4_N(BF_4_)) salt (TBATFB, from Sigma Aldrich, St. Louis, MO, United States), and 2 wt.% photochromic dichroic dye molecule ethyl 8-((4’-pentylcyclohexylphenyl)-difluoromethylphenyl-4-yl)-2-phenyl-2-(4-pyrrolidinyl phenyl)-2H-naphtho[1,2-b]pyran-5-carboxylate (from ITRI, Hsinchu, Taiwan), the chemical structure presented in Figure 2. All of the mixture was sandwiched between two glass substrates without any alignment treatment; the inner surfaces of both substrates were pre-coated with indium tin oxide to cause them to function as transparent conductive layers. The cell gap was approximately 12 μm.

Spectra were obtained using a USB4000 spectrometer (from Ocean Optics, Largo, FL, USA). The source of the incident light was a parallel tungsten halogen light HL-2000 (from Ocean Optics). The haze was measured using a haze meter NDH2000 (from Nippon Denshoku, Tokyo, Japan).

### 2.2. Driving Method and Principle

The smart window herein can be made to enter four optical states by exposure to UV light and the application of various voltages. Figure 3 shows a schematic switching diagram. In the absence of UV light, the sample is initially in the planar state, in which the liquid crystal molecules are all parallel to the substrate; the reflection wavelength of this cholesteric liquid crystal is modulated in the UV light region, so it is transparent in the visible light region, as shown in Figure 3a. Since the salt TBATFB is doped into the CLC, it disassociates into positive and negative ions and changes the electric conductivity of the liquid crystal. When a low-frequency voltage is applied to an LC with negative dielectric anisotropy (Δε < 0) and positive conductivity anisotropy (Δσ > 0), the turbulence effect can be observed. The ions will move parallel to the electric field [24]. The motion of the ions creates turbulence and tends to align the liquid crystal along the direction of movement. This turbulence effect distorts the helical structure in the planar state relative to that in the focal conic state. When the electric field is removed, the cholesteric liquid crystal remains in the focal conic state. The focal conic state displayed scattering because of its multi-domain structure, as shown in Figure 3b. To switch the CLC from the focal conic state to the planar state, a high-frequency voltage is applied. Because of the limited mobility of the ions, the high-frequency electric field cannot generate turbulence. Owing to the negative dielectric anisotropy of the cholesteric liquid crystal, the electric field tends to align the liquid crystal molecules perpendicular to the electric field. The helical twisting power that is generated by the chirality causes the CLC to enter the planar state. Therefore, the CLC can be switched between the planar and focal conic states by applying high-frequency and low-frequency pulse electric fields. Both of these states are stable at 0 V, in which the material is transparent and scattering, respectively.

The smart window is colorless when it is not irradiated by UV, but rapidly darkens when it is irradiated by UV. Photochromic dichroic dye naphthopyran-based materials undergo photo-induced conformational changes in molecular structure from closed form to open form. The structure of the dichroic dye is elongated and planar in its open form. The absorbance spectra of dichroic dye undergoes a red-shift from ultraviolet to the visible range upon UV light irradiation. Photo-induced conformational changes are short-lived and reversible. When the UV light source is removed, the material spontaneously returns to its closed, colorless form in a few minutes. Since the helical structure in the planar state has the helical axes perpendicular to the substrate, incident light of any polarization will be strongly absorbed by the photochromic dichroic dye, reducing transmittance; this state is the dark clear state, as shown in Figure 3c. Similarly, in the dark clear state, a low-frequency pulse voltage can be applied to cause ionic turbulence and formed focal conic state. In this state, the absorption by the doped photochromic dichroic dye and multi-domain scattering characteristics cause the smart window to be dark and opaque; in this state, the smart window has the lowest transmittance, as shown in Figure 3d. The smart window becomes colorless after the UV irradiation has been removed for a short period.

## 3. Results

The degree of absorption of visible light by photochromic dichroic dye can be controlled by exposure to, or the absence of, UV light. Scattering can be controlled by applying voltage pulses of different frequencies. Therefore, this smart window could adopt four optical states—transparent, scattering, dark clear, and dark opaque; each state was stable at 0 V. Figure 4 shows the transmission spectra in these four states, which are compared in Table 1 with respect to average transmittance and haze. The transmittance in the transparent state was high in visible region and almost no scattering occurred; the average transmittance exceeded 83%, and the haze was less than 6%; in the scattering state, the haze was uniform; the average transmittance was only 32.2% and the haze as high as 70.4%. When the dye was irradiated for 1 min with UV light with a wavelength of 365 nm and an intensity of 10 mw/cm^2^, the dye photochromic dichroic transitioned from closed form to open form and absorbed. Since the absorption peak of the photochromic dye is around 560 nm, the transparent state had deep purple discoloration. The average transmittance of the dark clear state was 30.3%, and the haze was 8.3%, slightly higher than in the transparent state; in the dark clear state, the average transmittance was lowest at 14.2%, and the haze was highest at 74.9%.

Transmittance versus frequency of voltage pulse is shown in the Figure 5. The CLC was initially in the planar state with high transmittance, then a 60 V voltage pulse with different frequency was applied to measure the transmittance. When a low-frequency voltage was applied across the cell, the motion of the ions created turbulence and tended to align the liquid crystal along the direction of movement. This turbulence effect distorted the helical structure and turned the CLC from the focal conic state to the planar state, where we chose 200 Hz as the low-frequency voltage pulse. When the frequency increased, the turbulence became weak and the dielectric effect of LC dominated the alignment. Therefore, it maintained in the planar state with high transmittance. Here we chose 3000 Hz as the high-frequency voltage pulse.

The electro-optical characteristic of the smart window was measured before and after irradiation by UV light; a voltage pulse with a duration time of 3 s was applied and the transmission was measured 10 s after the end of the pulse. Hence, all transmissions were measured at 0 V. Figure 6 plots the transmittance of the smart window after the removal of the applied voltage vs. the amplitude of the applied voltage pulse. A parallel tungsten halogen was the light source and a spectrometer was used as the detector. Figure 6a plots the T–V curve of the smart window in the absence of UV light. The transparent state switched into the scattering state as shown in Figure 6a (black line). Voltage pulses with a frequency of 200 Hz switched the liquid crystal from the transparent planar state to the scattering focal conic state. When the voltage pulse was below 40 V, the turbulence that was generated by the ions was not enough to cause a transition of the liquid crystal; when a voltage pulse over 40 V was applied, the turbulence effect was against the dielectric interaction because of the negative dielectric anisotropy of the liquid crystal; the helical structure in the planar state was disturbed, and ultimately switched to the focal conic state. As the voltage was increased, the turbulence effect became stronger, and the focal conic domain size became smaller, close to the wavelength of visible light. Therefore, more scattering reduced transmittance. When a voltage pulse of 50 V was applied, the transmittance was at its minimum, and the ionic turbulence was greater than the dielectric interaction. The scattering then switched into the transparent state as shown in Figure 6a (red line). Voltage pulses with a frequency of 3000 Hz switched the liquid crystal from the focal conic into the planar state. Pulses of 50 V and a frequency of 200 Hz were applied to reset the sample into the focal conic state before applied measurement voltage pulses. A voltage pulse with a duration time of 3 s was applied and the transmission was measured 10 s after the end of the pulse. Voltage pulses with a frequency of 3000 Hz yielded very weak ionic turbulence because the high-frequency voltage limited their mobility, so only negative dielectric anisotropy contributed to all of the electrical properties. When the voltage pulse was less than 20 V, the liquid crystal molecules were not arranged in parallel as the substrate, and the focal conic state was maintained. As the applied voltage pulse increased, some regions adopted the parallel substrate alignment due to the negative dielectric anisotropy of liquid crystal molecules and the formation of a planar structure as a result of the helical twisting power that arose from the chiral doping. When the voltage pulse exceeded 70 V, all liquid crystals were arranged in a planar state as a result of the electric field, and this state was stably maintained after the voltage was turned off.

Figure 6b plots the T–V curve of the smart window that was exposed to UV light. The sample was exposed for 1 min to UV light with a wavelength of 365 nm and an intensity 10 mw/cm^2^ before the transmittance was measured as a function of the amplitude of the voltage pulse. Irradiation by UV light changed the sample from colorless to deep purple. Owing to the structure in the planar state and the dichroic absorption of the photochromic dye, the transmittance decreased to about 30%. The focal conic state was dark and hazy with a lower transmittance of 14.2%. Switching completely from the dark clear state to the dark opaque state required a 52 V voltage pulse with the frequency of 200 Hz. Switching from the dark opaque state to the dark clear state required a 70 V voltage pulse with a frequency of 3000 Hz.

Table 2 compares the response time before and after irradiated by UV light. A 52 V voltage pulse with a frequency of 200 Hz was used to switch from the planar state to the focal conic state before and after irradiation by UV light. The response times before and after the color changed were 292 ms and 277 ms, respectively. A 70 V voltage pulse with a frequency of 3000 Hz changed the focal conic state to the planar state. The response times before and after the color changed were 16.8 ms and 23.6 ms, respectively. When the smart window was switched from the transparent state to the scattering state, the competition between the turbulence effect and the negative dielectric anisotropy of the liquid crystal caused the response time to be much longer than that associated with switching from the scattering state to the transparent state. Based on the above results of the experiments on the electro-optical characteristics, both in the presence and the absence of UV light, the photochromic dichroic dye only affects the optical properties, whereas the electrical properties do not significantly change.

Figure 7 plots the time-dependence of the optical response to exposure UV light. The smart window in the planar state was irradiated by UV light with a wavelength of 365 nm and an intensity of 10 mw/cm^2^ for 42 s, and thus changed from colorless to deep purple. The response time for the normalized transmittance to decline from 90% to 10% was 5.5 s. The UV light was then turned off and the light transmittance of the smart window returned to high. The response time for the normalized transmittance to increase from 10% to 90% was 53.9 s.

Figure 8 presents photographs of the proposed smart window. Each state existed stably without the application of a field. Figure 8a,b show the transparent state and scattering state without exposure to UV light. The transparent state is colorless and allows the background to be seen clearly. The scattering state is white and blocks the background. Figure 8c,d present the dark clear state and the dark opaque state under irradiation by UV light. In the dark clear state, since the absorption peak of the photochromic dye is around 560 nm, the sample has deep purple discoloration. The dark opaque state exhibits absorption and scattering, and so has the lowest transmittance and provides the best shielding. Figure 9 shows microscopic images of each optical states.

Figure 10 shows photographs of this smart window irradiated by real sunlight at noon at National Sun Yat-sen University. Before it is exposed to sunlight, the sample is in the transparent planar state and colorless, as displayed in Figure 10a. Figure 10b shows this smart window covered by a checkerboard mask and exposed to sunlight for five minutes. The part that was irradiated by sunlight entered the dark clear state. Finally, the mask was removed and the sample was exposed to sunlight for five minutes; a uniform dark clear state was obtained across the entire surface, as shown in Figure 10c.

## 4. Conclusions

In summary, this work demonstrated a photochromic dye and salt-doped cholesteric liquid crystal smart window that has four different optical states—transparent, scattering, dark clear, and dark opaque. The smart window can be switched from the transparent state (planar state) to the scattering state (focal conic state) by applying a 52 V voltage pulse with a frequency of 200 Hz, causing the transmittance to decrease from 83.35% to 32.3%; it can be switched back to the transparent planar state by applying a 70 V voltage pulse with a frequency of 3000 Hz, each state is stable over 48 hours or even longer in the absence of an applied voltage. Exposure to natural sunlight or artificial UV light changes its chromatic properties. The dimming transmittance is 30.3% and 14.2% in the dark and dark opaque states, respectively. After the UV source is removed, the smart window returns to its colorless state. Therefore, this smart window, which provides passive automatic dimming and active haze control for privacy protection, has great practical potential.

## Figures and Tables

**Figure 1 materials-13-04137-f001:**
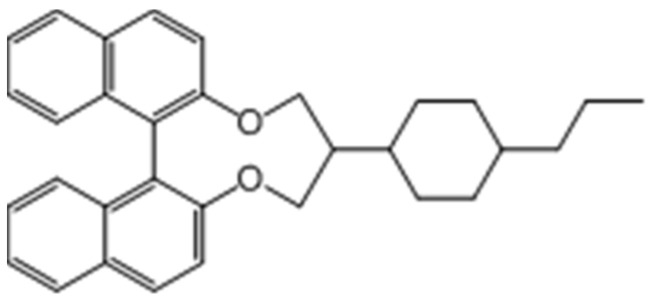
Chemical structure of chiral agent R-5011.

**Figure 2 materials-13-04137-f002:**
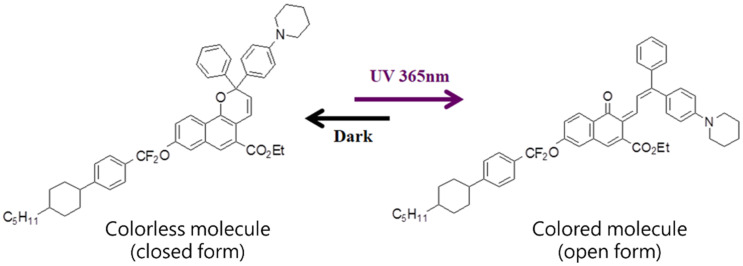
Chemical structure of photochromic dichroic dye ethyl 8-((4’-pentylcyclohexylphenyl)-difluoromethylphenyl-4-yl)-2-phenyl-2-(4-pyrrolidinyl phenyl)-2H-naphtho[1,2-b]pyran-5-carboxylate.

**Figure 3 materials-13-04137-f003:**
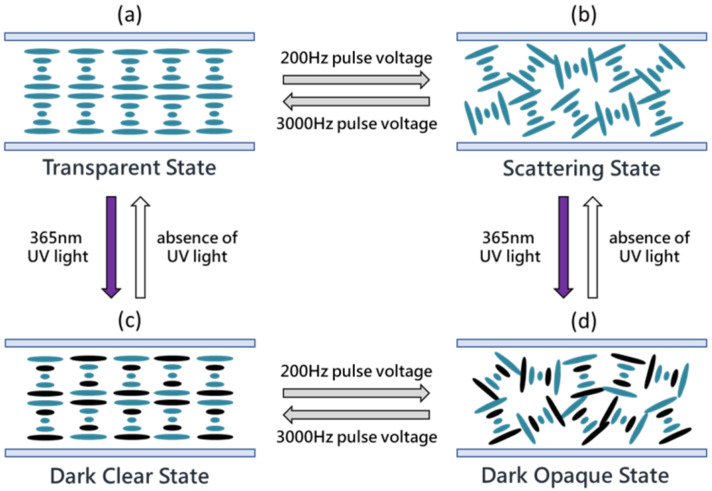
Schematic diagram and operation of photo- and electrical-controllable smart window. (**a**) Transparent planar state in visible region. (**b**) Scattering focal conic state. (**c**) Dimming (absorption) planar state. (**d**) Absorption scattering focal conic state.

**Figure 4 materials-13-04137-f004:**
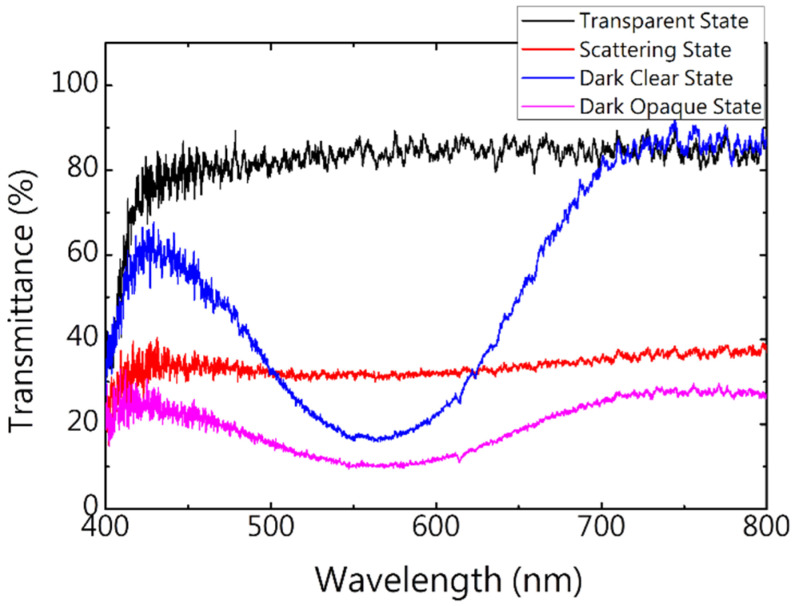
Transmission spectra in different optical states.

**Figure 5 materials-13-04137-f005:**
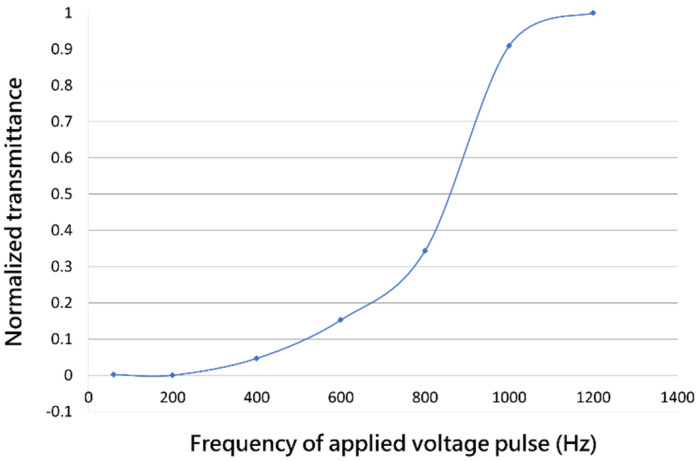
Normalized transmission as a function of the frequency of applied voltage (60 V).

**Figure 6 materials-13-04137-f006:**
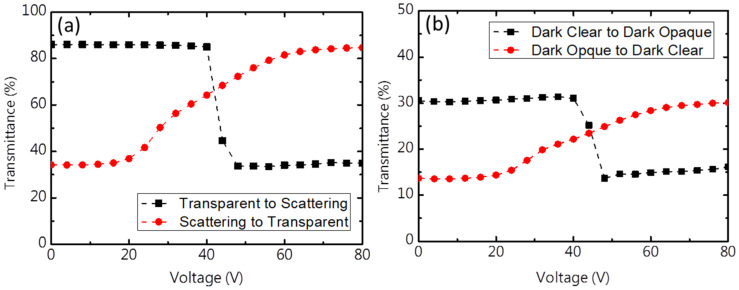
Transmittance versus amplitude of voltage pulse (**a**) before irradiation by UV light and (**b**) after irradiation by UV light.

**Figure 7 materials-13-04137-f007:**
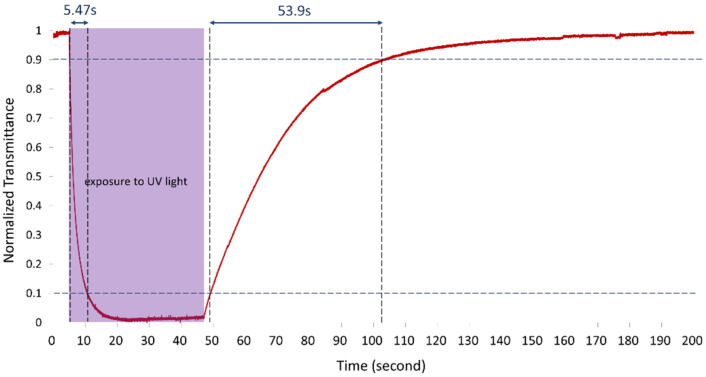
Optical response time under irradiation by UV light with wavelength of 365 nm and an intensity of 10 mw/cm^2^.

**Figure 8 materials-13-04137-f008:**
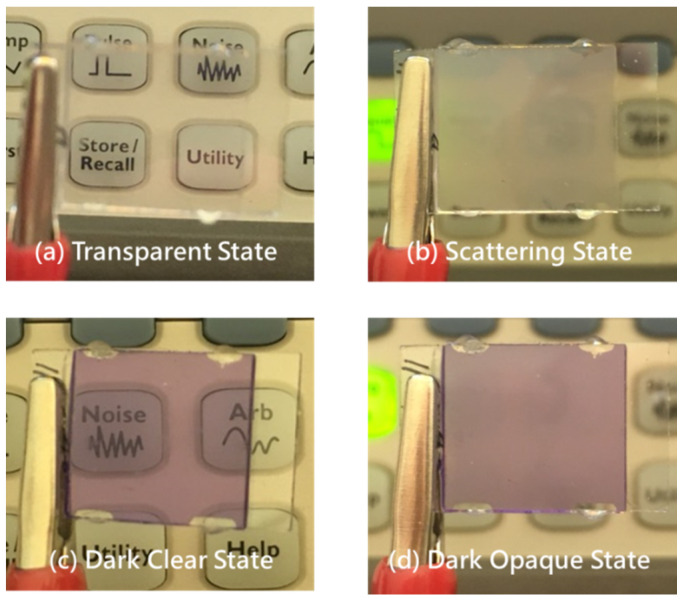
Photographs of (**a**) transparent state, (**b**) scattering state, (**c**) dark clear state, and (**d**) dark opaque state.

**Figure 9 materials-13-04137-f009:**
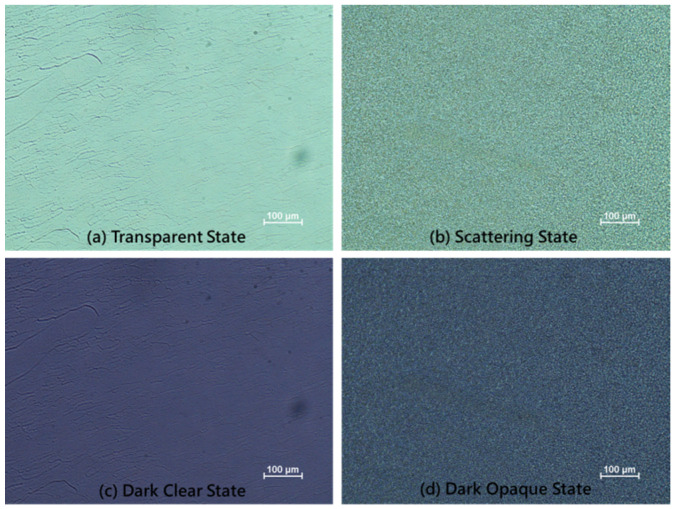
Microscopic images of (**a**) transparent state, (**b**) scattering state, (**c**) dark clear state, and (**d**) dark opaque state.

**Figure 10 materials-13-04137-f010:**
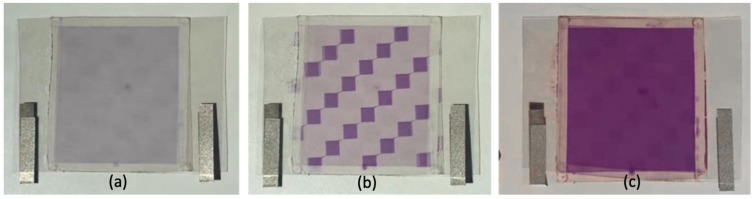
Photographs of this smart window (**a**) in planar state before exposure to sunlight, (**b**) covered by a checkerboard mask and exposed to sunlight for five minutes, (**c**) after the removal of the mask and exposed to sunlight for five minutes.

**Table 1 materials-13-04137-t001:** Comparisons of optical states.

Optical Properties	Transparent State	Scattering State	Dark Clear State	Dark Opaque State
Average Transmittance (%)	83.35	32.3	30.3	14.2
Haze (%)	5.9	70.4	8.3	74.9

**Table 2 materials-13-04137-t002:** Comparisons of response time before and after irradiation by UV light.

Response Time	Form Transparent to Scattering	Form Scattering to Transparent
Before Color Change (ms)	292	16.8
After Color Change (ms)	277.2	23.6

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
