# Peer review of "Smart Window with Active-Passive Hybrid Control"

_materials, 2020, doi:10.3390/ma13184137_

Round 1

Reviewer 1 Report

The paper presents the combination of two known features, i.e. the darkening of a photosensitive layer upon exposure to UV light, and the response of cholesterol liquid crystals to voltage pulses. Some optimisation was certainly done in the present work, but not disclosed in the manuscript. The authors stress the usefulness of the presented device as a smart window material, but they fail to compare it to present day solutions (curtains, blades etc.) especially with respect to feasibility, costs and material consumption of a full size device.

The authors should give structures (or references/CAS numbers) for the CLC ingredients DNM-9528 and R-5011.

The text includes some typos and/or badly chosen words which should be corrected: 

-line 55: present 'no' and 'ne' as 'n0' and 'ne'

-line 82: replace 'form' by 'from'

-lines 84/85: remove 'whose chemical structure is presented in Fig. 1', since the correct name of this simple chemical is given, which largely suffices.

-lines 94/95: hence, remove Fig. 1

-line 144: two consecutive commas instead of one

-line 159: 'time interval' presumably should mean 'duration'. Time interval could denote the time between two pulses, but then the indication of the pulse length would be missing.

Reviewer 2 Report

I recommend the manuscript by Heng-Yi Tseng et al. entitled “Smart Window with Active Passive Hybrid Control” to publish after minor revision in Materials. The manuscript fits well within the scope of the journal. In the reviewed text authors present electrically controlled transmission of the liquid crystal cell based on the well-know the dynamic scattering effect. By changing the frequency of applied electric field and applying UV light they obtained a transparent state, scattering state, dark opaque state, and dark clear state. The obtained results are interesting, worth publishing, and confirming the conclusions.

My specific comments are listed below. Responses should be included in the revised manuscript.

  1. Why authors choose the selected frequencies? The authors should show the dependence of transmission as a function of frequency for measuring cell and explain their choice of frequency.
  2. Does the scattering state is better with or without an applied electric field?
  3. What was the electrical conductivity of the investigated material? It is positive or negative?
  4. How long each state is stable after switching of the electric field? Did the authors check it?
  5. Line 55: no and ne should be subscripted.
  6. English spelling and grammar should be carefully checked.

Reviewer 3 Report

The article is interesting and the results have big potential of the application, but before publishing some corrections should be made:

1. The abstract is written in a general way. Specific values should be added.
2. the Introduction section should be extended to show the novelty of conducted research on the basis of the papers from the last two years.
3. line 89 can ITO be replaced with another conductive material?
4. How was the transmittance measured?
5. line 168 and 173: the authors have written " the turbulence effect was greater than the negative dielectric anisotropy of the liquid crystal" and "the ionic turbulence counterbalanced the negative dielectric anisotropy" - please clarify it
6. How can the tested material be placed on the window?
7. Is it possible to add some microscopic images of the samples?

Round 2

Reviewer 3 Report

In my opinion, the article is ready to be published